# The lived experiences and caring needs of women diagnosed with cervical cancer: A qualitative study in Dar es Salaam, Tanzania

**Emmanuel Z. Chona**[1]*, **Emanueli Amosi Msengi**[1], **Rashid A. Gosse**[2], **Joel S. Ambikile**[3]

1 Muhimbili University of Health and Allied Sciences, Dar es Salaam, Tanzania, 2 Department of Clinical Nursing Services, Muhimbili National Hospital, Dar es Salaam, Tanzania, 3 Department of Clinical Nursing, Muhimbili University of Health and Allied Sciences, Dar es Salaam, Tanzania

* emmanuelchona20@gmail.com

**Data Availability Statement:** All relevant data are within the paper and its Supporting Information files.

## Abstract

### Background

Cervical cancer continues to be a major global public health concern affecting the lives of many women and resulting in financial burdens. In 2020, cervical cancer was the seventh most commonly diagnosed cancer among all cancers worldwide and Tanzania was ranked fourth among the countries with the highest incidence rates (59.1 new cases per 100,000 women) of cervical cancer. The lived experience and caring needs of patients and their families provide insights into the psychosocial aspects of healthcare among the affected population. However, there is inadequate information concerning the lived experiences of cervical cancer patients in Tanzania and Sub-Saharan Africa in general. This study aimed to explore the lived experiences and caring needs of cervical cancer patients at Ocean Road Cancer Institute (ORCI) in Dar es Salaam, Tanzania.

### Methods

A qualitative descriptive study was carried out among cervical cancer patients at ORCI in Dar es Salaam, Tanzania. Using a purposeful sampling technique, 12 cervical cancer patients were interviewed with the principles of saturation guiding sample size determination. A semi-structured face-to-face interview guide was employed to collect the information. A conventional content analysis approach was used to analyze data after translation with the aid of NVivo 12.0 computer software.

### Results

Five themes emerged after data analysis: knowledge and attitude about cervical cancer, sufferings from a disease process, socio-economic disruptions, psychological problems, and sexual and reproductive concerns.

**Funding:** The author(s) received no specific funding for this work.

**Competing interests:** The authors have declared that no competing interests exist.

## Conclusion

The findings of this study provide insights into the life experiences and caring needs of cervical cancer patients and call for response from healthcare stakeholders to develop and implement comprehensive and culturally consonant approaches in providing care to the affected population. More qualitative studies are required to ascertain the lived experiences of advanced cervical cancer patients and those of long-term cervical cancer survivors.

## Introduction

Cervical cancer continues to be a major global public health concern and the leading cause of morbidity, mortality, and resulting in financial burdens worldwide [1, 2]. In 2020, the estimated number of women who were diagnosed and died due to cervical cancer was 604,000 and 342,000 respectively, whereby about 90% were from low-and middle-income countries [3]. Africa accounts for 20% of the new global cervical cancer diagnoses annually, with about 120, 000 new cases [4]. Also, the burden of cervical cancer continues to be substantial in Sub-Saharan African (SSA) countries which contribute to 24.55% of global deaths [5]. Tanzania is among Sub-Saharan African countries with 4th highest incidence rate of 59.1 new cases per 100,000 women in the world. The country has a high mortality rate of cervical cancer, with 42.7 deaths per 100,000 women [6]. The high burden of cervical cancer in LMICs is associated with limited access to public health services and poor implementation of screening, and treatment of this disease [4, 7, 8]. The World Health Organization (WHO) developed a global strategic initiative to eliminate the burden of cervical cancer by 2030 through vaccination, regular screening and prompt treatment of identified cases [3, 9]. The Ministry of Health in Tanzania has adopted the WHO global strategic initiative to develop an intervention plan to prevent, eliminate and manage cervical cancer cases [10]. By 2024, Tanzania planned to have a target vaccination coverage rate and target screening coverage rate of 85% and 35% respectively [6].

Lived experience with cervical cancer involves knowledge or experiences of a person with cervical cancer for those who are exposed to this medical condition [11]. Good lived experiences with different medical conditions are associated with being knowledgeable about the causes, signs, symptoms, and treatments of the respective diseases [12]. Patients with good lived experiences with chronic illnesses are more likely to have good health-seeking behaviors and ultimately maintain their health statuses [13, 14]. Women diagnosed with cervical cancer may experience a reaction to diagnostic results, isolation from community and family members, marriage break-ups, anxiety, sadness, and deterioration of health status [14, 16]. Moreover, they experience stigmatization from community members because most community members had misconceptions about the disease [17, 18], with significant others being physically isolated and not invited to various community events [19, 20].

Women with cervical cancer experience dissatisfaction with healthcare services, inadequate information from healthcare providers, and poor compliance with treatment regimens [21, 22]. Studies conducted in Ghana and Ethiopia revealed that women with cervical cancer experienced health deterioration caused by illness, psychological disturbance after receiving cancer diagnostic results, and experiencing devastating symptoms of the disease [15, 21]. Other women experienced economic challenges (loss of working ability, loss of employment, and high medical costs) and disruption of social relationships (marriage break-up, isolation by family and community members) [20, 23].

Despite the significance of understanding the lived experiences of cervical cancer patients in identifying the gap in health care among the affected population for better treatment and reduction of cervical cancer-related mortality, very few studies have been done so far, especially in Sub-Saharan African countries. This indicates a need to conduct more studies to get adequate information that will help key stakeholders in health care to develop proper strategies to address this problem. Therefore, this study aimed to explore the lived experiences and caring needs of cervical cancer patients at Ocean Road Cancer Institute in Dar es Salaam, Tanzania.

## Materials and methods

### Study design and setting

A qualitative descriptive study design was employed to explore the lived experiences and caring needs of cervical cancer patients at ORCI in Dar es Salaam, Tanzania. This design was employed because it allows an in-depth exploration of participants' experiences as proposed by Doyle et al., [24]. The ORCI is a national referral public center for cancer treatment and approximately 5,500 new patients with cancer are attended annually, of which approximately 39% have cervical cancer. The hospital is staffed by many specialized health professionals from several disciplines and receives all cancer patients referred from all regions in the country. The center provides chemotherapy, radiation therapy, complaint therapy, and other supportive and palliative care. It is the major area for cancer registry, early detection, prevention, standard treatment, and palliative care in Tanzania and it is the only cancer center in Dar es Salaam.

### Study population and eligibility criteria

This study involved a population of women diagnosed with cervical cancer at ORCI in Dar es Salaam, Tanzania. All women with cervical cancer for at least 1 year since diagnosis at the Institute were included in this study. Critically ill cervical cancer patients were excluded from this study because they could not be comfortable in responding to the questions due to potential life-threatening physiological conditions requiring critical care. Also, all cervical cancer patients who had cognitive impairment were excluded from this study because they could not provide the required information.

### Sampling procedure and sample size

A purposeful sampling technique was used to recruit the study participants. Records of patients were reviewed at the registry point to identify those who had been diagnosed with the disease. Cervical cancer patients who were able and willing to provide descriptions of their experiences and needs with the disease were purposively selected from those who met the inclusion criteria. A total of 12 early-stage cervical cancer patients were interviewed with sample size determination based on the principles of saturation as proposed by Malterud et al., [25], i.e.; sampling was terminated when no new information was obtained from the study participants.

### Data collection tool

In this study, a semi-structured interview guide developed based on reviewed literature and modified to suit the specific objectives of the study was used to collect information from the study participants. The guide was employed to gather in-depth information about the lived experiences and caring needs of women diagnosed with cervical cancer. A background information sheet was used to collect socio-demographic information from the study participants.

The interview guides were developed in English and then translated into Swahili, the national language to enhance better understanding by the study participants. Furthermore, it was pre-tested before the data collection was carried out to ensure the appropriateness of the questions.

## Data collection procedure

The interviews were conducted from December 2022 to February 2023 by the principal investigators (EZC, EAM and RAG) of this study with the assistance of 2 trained research assistants who had completed 2 days of intensive training at the data collection site. The interviews were carried out in a room providing privacy and comfort at the hospital premises to avoid external distractions. In all interviews, each question was followed by several probes to elicit more information or clarification of the responses provided by participants. Data collection was conducted in Swahili (the national language), and all interviews were digitally audio-recorded. Data that were not easily captured by the audio recorder, such as non-verbal cues were written down as field notes in the notebook by the research assistants. The duration of in-depth interviews lasted between 35 to 50 minutes.

## Data analysis

Data analysis was initiated soon after the first interview by transcription of the audio-recorded data by typing directly into the computer with the aid of the Microsoft Word program and then translated into the English language. Iterative reading of the transcripts was done by principal investigators separately with an open mind to obtain a general impression and an overall understanding of each transcript. Different colors were used to highlight the patterns in the text corresponding to the preconceived category stated in the study objectives. The transcribed and translated data sets were transferred to NVivo 12.0 computer software which is designed to help in organizing the qualitative data and coding the text. The conventional content analysis approach was employed to enable a deeper understanding and formation of themes after several reading iterations of the transcripts as proposed by Hsieh & Shannon [26]. Categories were developed from actual phrases in the text segments and sub-themes were extracted from each category by the principal investigators. Similar categories were linked to form themes (*Fig 1*). Field notes were also analyzed separately whereby the patterns and categories were compared to those from in-depth interviews. Data in the form of direct quotes were used to substantiate the relevant categories. To ensure the credibility of the findings, the authors discussed meanings emerging from the analysis outputs, categories, and themes according to the specific objectives of this study. The authors also held a meeting with interviewed participants to discuss and reach a consensus on the themes and categories identified. Their suggestions were obtained regarding the comprehensiveness of the findings and changes were incorporated accordingly.

## Ethical considerations

The ethical approval for this study was obtained from the Muhimbili University of Health and Allied Sciences (MUHAS) Institutional Review Board with a Ref. No. DA.282/298/01.C/1466. Permission to conduct the study was obtained from the ORCI administration with a Ref. No. 10/VOL.XXI111-B. Written informed consent was sought and obtained from all participants before the actual data collection procedure. Confidentiality of participants' information was strictly maintained. To further maintain confidentiality, only numbers were used to identify participants, and the audio-recorded interview transcripts were kept in a safe place and will be destroyed later after the completion of the study. Participation was fully voluntary, and the participants were informed of their full right to skip or ignore any question or withdraw their participation from the study at any stage.

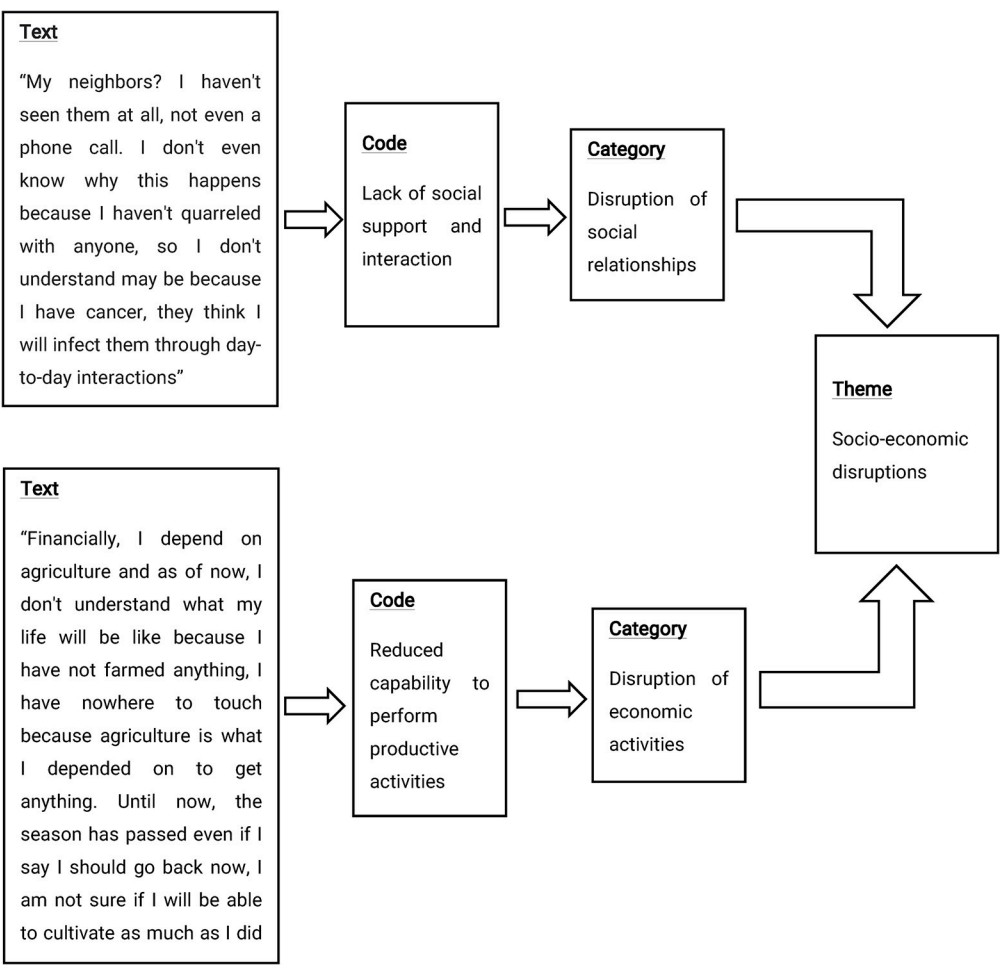

**Fig 1. The coding process and formation of themes.**

## Results

### Socio-demographic characteristics of study participants

A total of twelve participants participated in this study. The mean (±SD) age of participants was 53.3 (±14.5) years with half of the participants in the age group of 31 to 50 years. At the time of the interview, 9 participants had two years since being diagnosed with cervical cancer. Four participants were single, the other 4 were divorced, and 10 were Christians. Most (10 participants) had primary education and only one participant was formally employed. Half of the participants had a parity of 1 to 2 children and 9 had a stage II cancer diagnosis. Ten participants received concurrent radiotherapy and chemotherapy treatments. All participants had curative treatment intentions (*Table 1*).

### Themes identified

Five predominant themes emerged after data analysis as follows: knowledge and attitude about cervical cancer, sufferings from a disease process, socio-economic disruptions, psychological problems, and sexual and reproductive concerns (*Table 2*).

**Table 1. Socio-demographic characteristics of study participants.**

| Variables | Frequencies (n) |
|---|---|
| **Age in years** | |
| 31–50 | 6 |
| 51–70 | 3 |
| 71–90 | 3 |
| **Duration of cervical cancer since diagnosis** | |
| One year | 2 |
| Two years | 9 |
| Three years and above | 1 |
| **Marital status** | |
| Single | 4 |
| Married | 2 |
| Widow/widower | 2 |
| Divorced | 4 |
| **Religion** | |
| Christian | 10 |
| Muslim | 2 |
| **Level of education** | |
| No formal education | 2 |
| Primary education | 10 |
| **Occupation** | |
| Formally employed | 1 |
| Self-employed | 8 |
| Unemployed | 3 |
| **Parity** | |
| 0 | 1 |
| 1–2 | 6 |
| 3–5 | 2 |
| 6 and above | 3 |
| **Current stage of cancer** | |
| Stage I | 3 |
| Stage II | 9 |
| **Treatment intent** | |
| Curative | 12 |
| Palliative | 0 |
| **Radiotherapy concurrent with chemotherapy** | |
| Yes | 10 |
| No | 2 |

**Knowledge and attitude about cervical cancer.** Most participants reported having insufficient knowledge among themselves and the community at large about cervical cancer from etiology, the population at risk, signs and symptoms up treatment options available particularly during their initial life with the disease (before being diagnosed). The latter compounded the negative attitude towards the disease in their respective communities. This theme is explained under two categories; knowledge gaps and negative attitudes toward cervical cancer.

*Knowledge gaps*. The expressions of most participants indicated the presence of knowledge gaps among them and the society at large. People in the community including family members

**Table 2. Summary of themes and categories.**

| Themes | Categories |
|---|---|
| **(i) Knowledge and attitude about cervical cancer** | Knowledge gaps |
| | Negative attitude towards cervical cancer |
| **(ii) Sufferings from a disease process** | Reactions to news of a diagnosis |
| | Physical symptoms sufferings |
| | Treatment side-effects |
| **(iii) Socio-economic disruptions** | Disruption of economic activities |
| | Disruption of social relationships |
| **(iv) Psychological problems** | Lack of emotional support |
| | Psychological torture from cancer-related stigma |
| **(v) Sexual and reproductive concerns** | Interference with sexual intercourse |
| | Fear of loss of fertility |

of patients with cervical cancer lack correct information related to the disease in terms of what causes the disease, initial signs and symptoms, and treatment modalities.

> *"No, there is no one there who understands that cancer can be cured, that is, for someone who has not been here. I do not believe that there is someone in the street or the village who can know if cancer is curable, because if you tell someone that you have cancer, he/she gets a big shock"* (Participant 2).

Other participants went far in their expressions blaming their spouses for transmitting the disease to them as they perceived cervical cancer to be a sexually transmitted disease. They thought that their husbands contracted cervical cancer from outside as they were unfaithful to their relationships and consequently transmitted the disease to them.

> *"I don't know why you men are killers like this, can you imagine I have been faithful to him for more than 4 years now but he didn't care. He cheated me and came back with this deadly disease. I can't trust a man anymore after all these tragedies"* (Participant 6).

*Negative attitude towards cervical cancer.* Lack of correct information regarding cervical cancer in the community geared the prevailing negative attitudes towards it. The family members and society at large associated cervical cancer with the impossibility of recovery and the ultimate death sentence. They thought that being diagnosed with the disease could be the end of their story in this world due to limited understanding and prevailing negative attitudes in the community about cervical cancer. Participants expressed that; most people in their societies believed that any cancer cannot be cured and that once someone is diagnosed with it, have to prepare for his/her impending death.

> *"There were changes, we were sad because they were built with the belief that cancer is incurable, so my mother and my children were very sad. Some of my other brothers were among the first people to disappoint me as they were saying and announcing that cancer is incurable. Moreover, they claimed that here at ORCI where I came for treatments, many people die, thus they believed I won't get back home alive also"* (Participant 1).

The negative attitudes concerning cervical cancer were reported to affect the whole process of treatment initiation and compliance. Some participants expressed that they faced resistance

from their family members in their decision to undergo treatment as they believed it was just wasting of resources for treating incurable cervical cancer.

> *"Also, other brothers think that this disease is not healing, that is, they think why should we give our money to someone who has already been infected with this disease. They do not believe that this treatment is curative, so they don't believe I will recover"* (Participant 10).

**Sufferings from a disease process.**   Participants reported experiencing sufferings associated with their condition from the time they started to manifest signs and symptoms of the disease up to the time they were receiving treatment at ORCI. Sufferings experienced are explained under three categories; reactions to news of the diagnosis, physical symptoms sufferings, and treatment side effects.

*Reactions to news of the diagnosis.* Participants expressed sufferings and grieving after being notified that they have cervical cancer. Many reported grief after the diagnosis as they believed cervical cancer could mark the end of their lives. They expressed anger, frustration, and disillusionment with the breaking news of the diagnosis as they had the prior belief that once someone has the disease, he/she cannot fight it off.

> *"I received the news that I was diagnosed with cervical cancer very badly and I was very sad. I was overwhelmed with thoughts until I lost consciousness. With such suffering I was experiencing, my family including my mother also disappointed me after revealing the diagnosis, they were saying that cancer can't be cured"* (Participant 7).

The news about the diagnosis not only affected cervical cancer patients but also other family members who were in a grieving situation as they believed their beloved ones with the disease were in danger of dying from the disease.

> "*When I told my mother about the news, she was sad and she stayed like this for a week, she doesn't drink and she left the house, I went out to search her and fortunately I caught her and urged her to return home. She started telling me "My soul is hurting my daughter, you are dying*" (Participant 12).

*Sufferings from physical symptoms.* Participants reported experiencing debilitating physical symptoms when they started to manifest them through the disease process. The most common physical symptom experienced by cervical cancer patients was per vagina bleeding. The symptom made some participants develop further complications related to blood loss like anemia.

> *"I was very shocked when I was at home because the blood was coming out too much, I was a bedridden person and the blood was not cutting at all. Until I came here, I'm thankful, I have done the treatments and nowadays the blood comes out a little"* (Participant 9).

*Treatment side-effects.* Participants expressed facing varying numbers and extent of side effects attributed to the treatments they were undergoing at the institute. They described the side effects as debilitating and affected the treatment schedule and protocols as some patients were subjected to changes in treatment modalities to suit the actual situation.

> *"When I started the radiation, it caused me nausea, it was causing problems. There were days when I couldn't eat, that is, I didn't want any kind of food, so I stayed until the evening without eating anything"* (Participant 3).

**Socio-economic disruptions.** Different aspects of social life and economic status among participants were affected as reported by most of them. The disruption was expressed under the following categories; disruption of economic activities and disruption of social relationships.

*Disruption of economic activities.* Disruption of economic activities was highlighted by participants as their daily productive routines were interrupted by their illness hence, reduced capacity to produce what they were initially used to. This made them depend on other family members in meeting their basic needs including treatment costs.

*"My personal life has been affected, when I was growing up I was looking for a job to earn an income. Right now, I'm stuck because I'm sick. Also, my child's life has been affected because my child does not have a job, he always does people's work. Up to now, my grandchildren have not gone to school since the school opened. He is struggling to find the money for treatment and his children. The grandchildren have been suspended from school because they have no shoes, they don't have the things needed at school"* (Participant 9).

Other participants went far in their explanations reporting that; despite their ability to engage in their usual daily activities, customers of their services were annoyed by the physical symptoms they experienced. Their customers disliked their products claiming poor hygiene associated with physical symptoms of cervical cancer like frequent bleeding.

*"In terms of the economy, there is a lot that has affected me because the situation I had made me unable to do my work with such success I used to get. In the evening I usually used to prepare porridge and send it to sell but when it reached this point, my customers were annoyed with the situation I was passing through as blood sometimes came out profusely and customers disliked my porridge as I was dirty. It means that I was unable to stay in the crowd and do those jobs. I was staying with my children who are students and were by then missing some of their basic needs"* (Participant 2).

*Disruption of social relationships.* In the context of family and social relationships, participants expressed their experiences of lack of support from close relatives as it was before the diagnosis of the disease (cervical cancer). Some family members disappeared in the crucial period when patients were undergoing treatment and required both financial and social support to cater to treatment costs and cope with treatment protocols and side effects.

*"After having this problem, my sister ran away from me, and until now I have no information about her. Before I shared this information about the disease, our relationship was good, but after I shared it with her, she became useless to me and until now I don't know where she is and even communication became a problem"* (Participant 3).

Some participants reported having a disruption in their family relationships and social life due to their decision to leave their homes and attend a cancer institute for treatment. This was geared up by poor perception of cervical cancer treatment among family members as they believed cancer is incurable and any attempt to undergo the treatment was like misuse of scanty family resources in treating something known to have no treatment.

*"My brothers have not been involved with my illness because they were against my decision to come here for treatment. They know that the drugs here are very expensive but they don't give any support just because I went against their wishes of not coming for treatments here. They*

*wanted to preserve the family resources as they believed the disease can't be cured and thus, they better not spend money catering for palliative care"* (Participant 4).

**Psychological problems.** The psychological impact reported by cervical cancer patients was predominantly related to discrimination and stigma associated with a negative attitude and low knowledge towards cervical cancer among family members and society in general. The psychological problems experienced by participants were categorized into lack of emotional support and psychological torture from cancer-related stigma.

*Lack of emotional support.* Participants verbalized feelings of emotional exhaustion due to a lack of family and social support. They expressed feelings of depression and anxiety as a result of altered normal social routines as some of their family and society members isolated them believing that they can also get hurt by acquiring the disease from the affected patient. The tendency led to a huge psychological impact on cervical cancer patients as they were emotionally unstable, stressed, and thought much about how their life is going to be.

*"I feel bad, very bad because they don't support me in this situation. . .that is, out of ten people in the family, only three are in contact with me. . .Now I don't know, maybe due to the difficulty of life, that's why they don't give me help, I don't know actually and I don't understand what is behind this tragedy, because others have good financial abilities, but I don't understand why they don't support me"* (Participant 3).

*Psychological torture from cancer-related stigma.* Participants reported facing the stigma associated with their diagnosis of cervical cancer. Participants thought of their impending death as it was believed that cancer is a deadly disease and cannot be cured. The stigma further imposed psychological torture among patients, particularly when they were yet to start treatment and psychological support from nurses and doctors at the institute.

*"Before I shift the settlement to here at the hospital, I used to attend the clinic from my relatives here in the city. They discriminated me a lot to the extent I had depression, for instance, when I went for a short call in the toilets, they immediately visited the place and cleaned it. They thought that I will infect them with the disease if we could share such facilities at home. Even food they saved mine separately as they believed the disease was contagious"* (Participant 10).

**Sexual and reproductive concerns.** Most of the participants reported some sexual and reproductive concerns because their husbands were reluctant to accept the situation and support them throughout the treatment and after treatment. This theme is elaborated under the categories of interference with sexual intercourse and fear of loss of fertility.

*Interference with sexual intercourse.* Many participants revealed the interference with sexual intercourse attributed to cervical cancer diagnosis and symptoms manifestation. They reported that their husbands were annoyed and unhappy with them from the time they were diagnosed with the disease up to the manifestations of the disease's physical symptoms like cervical bleeding. They expressed that their husbands were unable to enjoy sex as before and that made them (husbands) run away from their wives who had cervical cancer and initiate new relationships elsewhere.

*"When my partner saw that I was sick like this, he just left me because he believed I would not recover and satisfy his sexual desires as before. Being abused because of my illness I think was the reason for our family separation as he used to snub my concerns at the moment. He*

*already assumed that I will not be able to participate in sexual intercourse with him"* (Participant 5).

Other participants revealed the harsh treatment they experienced from their husbands just because they were uncomfortable with sexual intercourse. Despite them being in difficult illness situations, they have confiscated even the little assets they had and left empty-handed with no resources to support the treatments and for acquiring other basic needs.

*"I can say that my husband is a colonist because after I told him about my limited comfortability with sexual intercourse, he has taken everything from me. So, I am left with nothing. Even when I came here, he didn't give me anything, therefore, I couldn't even buy drinking water on the way"* (Participant 6).

*Fear of loss of fertility.* The diagnosis of cervical cancer among participants brought a hard-emotional impact related to their reproductive health. Being on chemotherapy and radiation therapy, participants revealed their doubts about the unknown and uncertainty of fertility prognosis after treatment completion. They were concerned about a loss of fertility as they perceived the disease itself and the treatments to be detrimental to the cervix.

*"I stopped sharing with my husband because I was really scared. I told him that it was for the sake of my life rather than giving birth later on, so I only did a little with him. He is a colonist, he doesn't realize that this person is sick, so sometimes I run away from him because I consider myself weak and no longer fertile"* (Participant 8).

## Discussions

The present study explored the lived experiences and caring needs of women impacted by cervical cancer at ORCI in Dar es Salaam, Tanzania. The mere fact that women diagnosed with cervical cancer in the current study did not have palliative treatment intentions did not mean they were immune to devastating life experiences imposed by the disease. Being women diagnosed with cervical cancer in a resource-limited setting and having a low community understanding of the disease would not be so kind and favorable to them. Hence, the findings of this study revealed various biological, psychological, social, and economic challenges that cervical cancer patients experienced from the time they were diagnosed with the disease and started to manifest symptoms of the disease.

The findings of this study revealed the low level of knowledge about cervical cancer in terms of what causes the disease, risk factors, signs and symptoms, and treatment options available among patients themselves and society in general. The presence of knowledge gaps concerning cervical cancer compounded the negative attitudes towards the disease which was very detrimental to the overall psychosocial well-being of the affected population. The low level of knowledge and unfavorable attitude regarding cervical cancer has been elucidated by several other studies [14, 15, 16]. Such findings also distort the screening programs among women, leading to late disease detection and consequently late consultation for orthodox health services. These findings illuminate the need for responsible authorities in developing and delivering mass education programs concerning cervical cancer with much emphasis on risk factors, signs and symptoms, and treatment modalities available to enhance early screening and rule out prevailing poor beliefs regarding the disease [27, 28].

Participants from this study reported experiencing suffering attributed to news about the diagnosis, disease process, and treatment-related effects. The fact that most participants

perceived cervical cancer as a death sentence, it was difficult for them to handle the news as a result they manifested maladaptive coping strategies like self-isolation. The news of the diagnosis affected other family members as they grieved for the impending death of their beloved ones. Consistent findings were reported from a study conducted in Japan among cervical cancer patients with terminal illnesses as they expressed experiencing extreme suffering and desired to at least die peacefully [18]. Also, this finding corroborates those of studies conducted among cervical cancer patients in Zambia and Ghana [14, 15]. Despite the participants from those studies suffering from the news of the diagnosis, the extent of suffering was quite different depending on the context of the study as settings with adequate knowledge of cervical cancer had minimal suffering because women knew the story behind the disease. In the current study, participants reported suffering from the physical symptoms of the disease and treatment side effects. Consistent findings were reported by adolescent cancer patients in studies conducted in Taiwan and Singapore [17, 29]. The most voiced symptom was bleeding that made women uncomfortable and some of them developed hematological deficits and nausea and vomiting were reported as debilitating for patients undertaking chemotherapy. More studies are needed to explore how better healthcare can be provided to relieve the sufferings associated with the symptom and ultimately promote healthier functioning for patients impacted by cervical cancer in such resource-limited settings [30].

The psychological problems experienced by cervical cancer patients in this study are similar to what was found in an exploratory study conducted in Ghana among cervical cancer patients where participants reported facing stressful experiences and depression after being frustrated with abusive interactions with some family members and society at large [15]. They were emotionally humiliated and psychologically tortured by cancer-related stigma due to the prevailing negative attitude towards cervical cancer in the community. Similar psychological distress was reported in a study done in Ethiopia among cervical cancer patients during follow-up care [21]. Hopelessness and lack of emotional support reported in the current study have also been reported by other studies done in different settings [14, 29, 31] where cancer was perceived as a death sentence for diagnosed patients. Public awareness programs are required to address such information gaps and rule out prevailing misconceptions concerning cervical cancer and other cancerous diseases in general [12]. Also, providing psychological and emotional support to women with cervical cancer as well as their caregivers is imperative for enhancing their capability among them to cope with the situation and lead a quality life. One of the ways to make this possible is by having cumulative efforts among healthcare providers and allied stakeholders in developing tangible strategies to address the concern [27].

Women with cervical cancer experienced disruptions in social and economic aspects of their life, which could be explained by a reduced capability to produce as they used to do and social isolation due to limited understanding and negative attitudes concerning cervical cancer. Also, the context itself from where patients were living was resource-limited by nature and thus the ability to acquire basic needs was dependent on their daily activities. So, having cervical cancer with such debilitating physical symptoms reduced their abilities to perform usual productive activities leading to an economic crisis among them and their families. The cost of treatments also rendered some patients' families bankrupt with others forced to sell the assets they had to cater for such costs. Similar findings of socioeconomic disruptions as a result of the disease were reported in other studies [14, 15, 21]. It was also revealed that the disruption imposed by cervical cancer on social and economic aspects of patients' life was a source of constant stress, low self-esteem, and isolation [14, 32]. This finding in the current study implies that healthcare providers should conduct a regular individualized assessment to explore patients' social and financial needs to ascertain the extent of socio-economic disruptions for early and proper linkage to social support groups [33].

The findings of this study also revealed the presence of sexual and reproductive concerns among women with cervical cancer. Husbands of women diagnosed with cervical cancer were reported to isolate their sick wives leading to family breakouts. Lack of knowledge about the disease among men geared up their decisions to abandon their wives as they perceived cervical cancer to have a detrimental effect on women's reproductive health by making them no longer fertile. Having such sexual and reproductive concerns is not limited to Tanzanian women in this study but is also revealed by women with cervical cancer in studies conducted in other Sub-Saharan African countries [14, 21]. A study conducted in Uganda among patients diagnosed with cervical cancer revealed that women experienced disruption in their sexual relationships including being deserted by their partners and family [16]. Women with cervical cancer revealed that they were scared of the unknown about their fertility particularly after undergoing radiotherapy and chemotherapy treatments. This may have negative psychological implications leading to a delayed recovery process and maladaptive coping behaviors among the affected women [34]. Comprehensive and holistic care should be given to women and their families in general to tackle the unmet needs among couples concerning sexual and reproductive concerns in relation to the disease itself, treatment modalities, and ultimate prognosis [35, 36]. However, other factors related to the sexual and reproductive health of women impacted by cervical cancer need further investigation.

## Study limitations

The limitation of this study is that all participants recruited had curative treatment intention, and there was no participant in the terminal stages of cancer. Therefore, the findings of this study might not reflect the lived experiences and caring needs of patients with terminal cervical cancer.

## Conclusion and recommendation

This study provides insights into the life experiences and caring needs of cervical cancer patients with broad bio-psychosocial perspectives. The findings of this study call for the development and implementation of comprehensive and culturally consonant approaches among health care providers and stakeholders in providing care to cervical cancer patients with more emphasis on psychosocial aspects of life as patients encounter huge psychological sufferings and disrupted social relations. At the health facility level, well-structured and sustained counseling sessions should be conducted to enhance cervical cancer patients' coping capability. Furthermore, qualitative studies should be done to explore the lived experiences and caring needs of advanced cervical cancer patients and those of long-term cervical cancer survivors to ascertain their experiences post-cancer treatment with special emphasis on psychosocial aspects of life.

## Supporting information

**S1 Text. A guide for in-depth interview.**
(DOCX)

**S2 Text. A STROBE checklist.**
(DOCX)

**S3 Text. Excerpts of transcripts.**
(DOCX)

## Acknowledgments

We are grateful to all participants for taking the time to participate in this study. Special appreciation goes to Annastasia Giles Mitema and Jennifer Joseph Mlingi, for organizing, contacting, and keeping appointments with study participants. We also thank the ORCI administration for permitting us to conduct the study at the institute.

## Author Contributions

**Conceptualization:** Emmanuel Z. Chona, Emanueli Amosi Msengi, Rashid A. Gosse, Joel S. Ambikile.

**Data curation:** Emmanuel Z. Chona, Emanueli Amosi Msengi, Rashid A. Gosse.

**Formal analysis:** Emmanuel Z. Chona, Emanueli Amosi Msengi, Rashid A. Gosse, Joel S. Ambikile.

**Investigation:** Emmanuel Z. Chona, Emanueli Amosi Msengi, Rashid A. Gosse.

**Methodology:** Emmanuel Z. Chona, Emanueli Amosi Msengi, Rashid A. Gosse, Joel S. Ambikile.

**Project administration:** Emmanuel Z. Chona, Emanueli Amosi Msengi, Rashid A. Gosse.

**Resources:** Emmanuel Z. Chona, Emanueli Amosi Msengi, Rashid A. Gosse.

**Software:** Emmanuel Z. Chona, Emanueli Amosi Msengi, Rashid A. Gosse.

**Supervision:** Joel S. Ambikile.

**Validation:** Emmanuel Z. Chona, Emanueli Amosi Msengi, Rashid A. Gosse, Joel S. Ambikile.

**Visualization:** Emmanuel Z. Chona, Emanueli Amosi Msengi, Rashid A. Gosse.

**Writing – original draft:** Emmanuel Z. Chona, Emanueli Amosi Msengi, Rashid A. Gosse.

**Writing – review & editing:** Emmanuel Z. Chona, Emanueli Amosi Msengi, Rashid A. Gosse, Joel S. Ambikile.

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
