## [Decision Letter · Decision Letter 0]

6 Jun 2023

PONE-D-23-10682The lived experiences and caring needs of women diagnosed with cervical cancer: A qualitative study in Dar es Salaam, TanzaniaPLOS ONE

Dear Dr. Chona,

Thank you for submitting your manuscript to PLOS ONE. After careful consideration, we feel that it has merit but does not fully meet PLOS ONE’s publication criteria as it currently stands. Therefore, we invite you to submit a revised version of the manuscript that addresses the points raised during the review process. Please see the reviewers' comments below and respond to each comment. In addition:- It should be noted in the methods that all study participants had early stage disease.- The background should be made more concise.- There is redundancy in the quotations, please remove redundant passages.- The English needs polishing.

We look forward to receiving your revised manuscript.

Kind regards,

Eric L Krakauer, MD, PhD

Academic Editor

PLOS ONE

Reviewers' comments:

Reviewer's Responses to Questions

**Comments to the Author**

1. Is the manuscript technically sound, and do the data support the conclusions?

Reviewer #1: Yes

Reviewer #2: Partly

2. Has the statistical analysis been performed appropriately and rigorously? 

Reviewer #1: N/A

Reviewer #2: N/A

3. Have the authors made all data underlying the findings in their manuscript fully available?

Reviewer #1: Yes

Reviewer #2: No

4. Is the manuscript presented in an intelligible fashion and written in standard English?

Reviewer #1: Yes

Reviewer #2: No

5. Review Comments to the Author

Reviewer #1: This is a lovely qualitative study that enriches the field. The authors quite rightly note that a limitation was not interviewing women with cervical cancer who have progressed and maybe dying. But I note that was part of their exclusion criteria. So the authors should add in their discussion that future studies should include this group

Reviewer #2: the paper reports the qualitative analysis done on 12 interviews of women with stages I-II cervical cancer treated at national referral public center for cancer in Tanzania .The objectives of the study were to explore the lived experiences and caring needs of the patients .The interviews followed a semistructured approach with 8 prespecified questions followed by probes during the interview.the methodology needs some clarifications .In the data analysis(line175) it is specified that some statements in the transcribed interviews were highlighted according to the preconceived categories stated in the study objectiveas s,which is nto the case,please clarify or report them.Question 7 exploring the experience with the health care services are not mentioned and it is not part of the themes and categories,explain why ..Line 184 To ensure the credibility..credibility is not a term used in qualitatve analysis,do the authors mean reliability? plese clarify The quotes taken from the interviews represent a significant part of the paper,there are overlappings and they make difficult the reading I suggest to reduce their number , to keep the signficant statements and report them in extent with references to the paper in the appendix

English to be fully and carefully revised

6. PLOS authors have the option to publish the peer review history of their article (what does this mean?). If published, this will include your full peer review and any attached files.

Reviewer #1: No

Reviewer #2: No

---

## [Author Response · Author response to Decision Letter 0]

19 Jul 2023

Editor's Comments

01. It should be noted in the methods that all study 

participants had early-stage disease

-We have added in the methods that 12 earlystage cervical cancer patients were recruited 

for the study.

Please see the revised manuscript (Line 125).

02. The background should be made more concise 

-The background has been revised to make it 

more concise.

Please see the revised manuscript.

03. There is redundancy in the quotations, please 

remove redundant passages

-We have revised the analysis critically and we 

removed redundant quotations reporting the 

same category in a specific theme.

Please see the revised manuscript.

04. The English needs polishing

The revised manuscript has been thoroughly 

copyedited by an expert university faculty 

competent in English with a strong scientific 

background in the field of medical research to 

make it clear and unambiguous.

Please see the revised manuscript.

Reviewer’s comments

01. Is the manuscript technically sound, and do the 

data support the conclusions?

The manuscript must describe a technically 

sound piece of scientific research with data 

that supports the conclusions. Experiments 

must have been conducted rigorously, with

appropriate controls, replication, and sample 

sizes. The conclusions must be drawn 

appropriately based on the data presented.

-After we reviewed all parts of our manuscripts 

followed the reviewer’s comments, we found 

some technical errors of our manuscript and 

we corrected them accordingly. We ensured 

that the conclusions made in the manuscript 

are drawn appropriately based on our study 

findings presented.

Please see the revised manuscript.

02. Have the authors made all data underlying the 

findings in their manuscript fully available?

-After we reviewed the PLOS Data policy, we 

have agreed to share our minimal anonymized 

data set (Excerpts of transcripts) as supporting 

information.

03. Is the manuscript presented in an intelligible 

fashion and written in standard English?

-The revised manuscript has been thoroughly 

copyedited by an expert university faculty 

competent in English with a strong scientific 

background in the field of medical research to 

make it clear and unambiguous.

Please see the revised manuscript.

04. The authors quite rightly note that a limitation 

was not interviewing women with cervical 

cancer who have progressed and maybe dying. So, the authors should add in their discussion 

that future studies should include this group. 

-We have revised the manuscript to incorporate 

the recommendation that further qualitative 

studies should be done to explore the lived 

experiences and caring needs of terminal 

cervical cancer patients.

Please see the revised manuscript (Line 503-

506).

05. In the data analysis (line 175) it is specified 

that some statements in the transcribed 

interviews were highlighted according to the 

preconceived categories stated in the study 

objectives, which is not the case, please clarify 

or report them

-During the analysis process, authors performed 

iterative reading of the transcribed data to 

obtain a general impression and an overall 

understanding of each transcript. With the 

objective of this study being exploration of the 

lived experiences and caring needs of cervical 

cancer patients, hence meaning units reflecting 

the study’s objectives were highlighted with 

different colors during iterations for easy 

generation of patterns and codes from the text. 

Hence, categories were developed from actual 

phrases in the text segments and similar ones 

were linked to generate themes.

Please see the revised manuscript (Line 151-

160).

06. Question 7 exploring the experience with 

health care services are not mentioned and it is 

not part of the themes and categories, explain 

why

-The experience with health care services were 

explored as per question 7. During the analysis 

process, several codes emerged concerning the 

health care services which were used to generate different categories reported in the 

manuscript. For instance, high costs associated 

with radiation therapy, high consultation fees, 

limited capability to purchase chemotherapy 

drugs were among the codes revealed by some 

participants under the aspect of health care 

services which were then linked with other 

codes arising from response to different 

questions like reduced capability to perform 

productive activities to generate a category of 

financial hardship. Also, codes like 

chemotherapy associated side-effects 

(example nausea and vomiting) were revealed 

upon probes on health care services which 

were then combined with other codes like 

radiotherapy associated side-effects to 

generate a category of treatment side-effects. 

Please see the revised manuscript.

07. Line 184, to ensure credibility. Credibility is 

not a term used in qualitative analysis, do the 

authors mean reliability? Please clarify

-By stating credibility, we meant the degree to 

which the results of a qualitative research are 

credible or believable from the participant’s 

perspectives since the purpose of qualitative is 

to describe or understand phenomena of interest from participants, hence they are the 

only ones who can legitimately judge the 

trustworthiness of the results. With this 

technique of establishing trustworthiness, data 

interpretations and conclusions were discussed 

by investigators and shared with the 

participants to allow them to clarify what their 

intentions were, correct misunderstandings 

and provide additional information if 

necessary.

08. The quotes taken from the interviews represent 

a significant part of the paper, there are 

overlapping and they make difficult the 

reading. I suggest to reduce their number, to 

keep the significant statements and report them 

in extent with references to the paper in the 

appendix

-We have revised the analysis critically and we 

removed redundant quotations reporting the 

same category in a specific theme.

Please see the revised manuscript.

09. English to be fully and carefully revised

-The revised manuscript has been thoroughly 

copyedited by an expert university faculty 

competent in English with a strong scientific 

background in the field of medical research to 

make it clear and unambiguous.

Please see the revised manuscript.

---

## [Editor Report · Decision Letter 1]

31 Jul 2023

The lived experiences and caring needs of women diagnosed with cervical cancer: A qualitative study in Dar es Salaam, Tanzania

PONE-D-23-10682R1

Dear Dr. Chona,

We’re pleased to inform you that your manuscript has been judged scientifically suitable for publication and will be formally accepted for publication once it meets all outstanding technical requirements.

Kind regards,

Eric L Krakauer, MD, PhD

Academic Editor

PLOS ONE

Additional Editor Comments (optional):

Your paper is publishable with some minor edits as follows:

LINE 25: Please change "increasing costs of treatment" to "resulting in financial burdens" (or clarify what is meant).

LINES 46 and 497: Delete the word "vital"

LINE 47: Delete "urgent"

LINE 50: Change "terminal" to "advanced"

LINES 83-84: Shorten to "Women with cervical cancer experience ... "

LINE 87: Delete "positive"

LINE 122: Delete "patients"
---

## [Editor Report · Acceptance letter]

2 Aug 2023

PONE-D-23-10682R1 

The lived experiences and caring needs of women diagnosed with cervical cancer: A qualitative study in Dar es Salaam, Tanzania 

Dear Dr. Chona:

I'm pleased to inform you that your manuscript has been deemed suitable for publication in PLOS ONE. Congratulations! Your manuscript is now with our production department. 

Kind regards, 

on behalf of

Dr. Eric L Krakauer 

Academic Editor

PLOS ONE